# Discovery and Characterisation of Novel Poly-Histidine-Poly-Glycine Peptides as Matrix Metalloproteinase Inhibitors

**DOI:** 10.3390/biom15050706

**Published:** 2025-05-12

**Authors:** He Wang, Wenchao Cai, Zhiyu Tang, Juanli Fu, Enrico König, Nanwen Zhang, Xiaole Chen, Tianbao Chen, Chris Shaw

**Affiliations:** 1School of Integrative Medicine, Fujian University of Traditional Chinese Medicine, Fuzhou 350000, China; 2013057@fjtcm.edu.cn (H.W.); 2231006013@fjtcm.edu.cn (J.F.); 2Department of Bioengineering and Biopharmaceutics, School of Pharmacy, Fujian Medical University, Fuzhou 350000, China; caiwenchao1@fjmu.edu.cn (W.C.); ztang2023@fau.edu (Z.T.); 3Laboratory of Synthetic and Structural Vaccinology, University of Trento, 38100 Trento, Italy; enrico.koenig@immaginabiotech.com; 4Medicine Natural Peptide Discovery Group, School of Pharmacy, Queen’s University Belfast, Belfast BT7 1NN, UK; t.chen@qub.ac.uk (T.C.); chris.shaw@qub.ac.uk (C.S.)

**Keywords:** snake venom, molecular cloning, matrix metalloproteinases, peptide inhibitor

## Abstract

For the first time, two poly-histidine-poly-glycine peptides (pHpG-H5 and pHpG-H7) were identified as promising candidates for matrix metalloproteinase inhibitors. cDNAs encoding pHpG-H5 and pHpG-H7 peptides were isolated from the *Atheris squamigera* cDNA library constructed using oligo(dT)-primed reverse transcription. Deduced sequences of pHpG peptides were systematically organised and utilised as templates for synthesising chemical replicates. All synthetic pHpG peptides exhibited inhibitory effects on human matrix metalloproteinase-1 (MMP-1). Spectroscopic analyses and molecular modelling demonstrated that pHpG peptides disrupt zinc ion coordination within the central catalytic domain of MMP-1, thereby inhibiting its enzymatic activity. As a novel peptide inhibitor of matrix metalloproteinase, pHpG-H7 modulates multiple biological processes, such as cell migration and angiogenesis, suggesting significant therapeutic potential.

## 1. Introduction

The variable bush viper, *Atheris squamigera*, is a snake species endemic to West and Central Africa. Although this viper is known to be venomous, the natural components of its venom are not yet fully understood. Currently, it is known that its snakebite causes local haemorrhage, thrombocytopenia, coagulopathy, and haemolysis [1,2]. A proteomic analysis of the venom was conducted in our laboratory, identifying phospholipases, disintegrins, serine proteases, and metalloproteinases as the major toxin components [3]. These components are likely to account for some of the symptoms resulting from envenomation.

In addition to these “classical” toxins, a novel class of peptides with clusters of histidine and glycine residues (poly-His-poly-Gly, pHpG) was first described in the venom of *A. squamigera* [4]. pHpG peptides exist in several molecular isoforms with varying numbers of repetitive glycine and histidine residues. The poly-histidine regions are frequently found in natural proteins and may contribute to the metal-chelating properties, while glycine residues may play a regulatory role [5,6]. Peptides with histidine and glycine residues probably act as modulators of proteinases by interacting non-covalently with the active site.

As important zinc-containing endopeptidases, metalloproteinases exhibit high homology in their conserved functional domains and activities, such as the mammalian matrix metalloproteinases (MMPs) and snake venom metalloproteinases (SVMPs) [7,8,9]. Since pHpGs have been identified as small-enzyme inhibitors targeting SVMPs in viper venoms [10], we hypothesise that the pHpG peptides of *A. squamigera* could possess MMP inhibition activity by coordinating the metal cofactor, which is crucial for substrate binding and enzymatic cleavage.

In this study, a transcriptomic investigation of lyophilised *Atheris squamigera* venom was conducted. The cDNA sequences of two poly-His-poly-Gly peptides, pHpG-H5 and pHpG-H7, were cloned and characterised. The chemical replicates of pHpG peptides were synthesised via solid-phase peptide synthesis and detected by a proteinase inhibition assay of MMP-1. A combination of a metal-chelation study, spectroscopic analyses, and 3D molecular modelling was employed to elucidate the inhibitory mechanism of pHpG peptides, underscoring their potential for the development of innovative therapeutic agents.

## 2. Materials and Methods

### 2.1. Molecular Cloning and Transcript Analysis

The lyophilised venom of *Atheris squamigera* was obtained from a commercial source (Latoxan, Portes-lès-Valence, France) and the cDNA library was constructed via the reverse transcription of mRNA isolated from glandular cells present in snake venom. Generally, five milligrams of venom were dissolved in the cell lysis/mRNA-stabilisation solution, followed by polyadenylated mRNA extraction using oligo(dT) Dynabeads (Dynal Biotech, Wirral, UK). Reverse transcription was performed using the manufacturer’s protocolto construct a cDNA library. The 3′-rapid amplification of cDNA ends (3′-RACE) was carried out to obtain full-length nucleic acid sequence data using a SMART-RACE kit (Clonetech, Oxford, UK). Generally, for this step, degenerate sense primers were designed based on the protein sequences reported in a previous study [4]. Additionally, a nested universal primer (NUP, 5′-AAGCATRGGTATCAACGCAGAGT-3′) supplied by the manufacturer was employed. The PCR products were purified, cloned using a pGEM^®^-T vector system (Promega, Madison, WI, USA), and sequenced by an ABI 3100 automated sequencer. The antisense primer was designed according to the sequence information obtained from 3′-RACE and was employed in the 5′-RACE to investigate the full-length sequence, including 3′-non-translated regions.

Chromas software (Version 1.45) was employed for DNA sequence analysis. Potential open-reading frames were identified using the ExPASy translation tool (www.expasy.org/tools/dna.html, accessed on 14 March 2023) and subjected to BLAST using the online tool of NCBI (http://blast.ncbi.nlm.nih.gov/Blast.cgi, accessed on 16 March 2023). Multi-sequence alignment was performed using Vector NTI software 11.5.3.

### 2.2. Peptide Synthesis and Purification

Following the establishment of the primary structure of natural pHpG peptides, replicates were synthesised using solid-phase Fmoc chemistry on a PS3 automated peptide synthesiser (Protein Technologies, Stockport, UK). The peptides were cleaved from the resins at the end of the synthesis, using 95%/2.5%/2.5% (*v*/*v*/*v*) trifluoroacetic acid/triisopropylsilane/water and purified by filtration and precipitation.

Synthetic peptides were purified by reverse-phased high-performance liquid chromatography (HPLC) using a linear gradient of 100% -0 buffer A (acetonitrile, 0–100% buffer B (acidified water) over 80 min. It is important to note that the solid-phase peptide synthesis of pHpG peptides was extremely difficult to perform and required several attempts to obtain the authentic product with an adequate degree of purity.

### 2.3. Mass Spectrometric Identification

Electrospray ionisation mass spectrometry (ESI-MS) and tandem mass spectrometry (MS/MS) analyses of synthetic pHpGs were conducted using a Q-TOF micro mass spectrometer (Waters, Milford, MA, USA). All peptides were detected in a standard ESI source at a flow rate of 0.2 mL/min. Spectrum were recorded in positive ion mode and analysed by using MassLynx 3.5 software (Waters, USA). Collision energy was manually optimised to maximise fragment ion coverage across the mass range.

### 2.4. Circular Dichroism (CD) Analysis

The synthetic pHpGs (0.2 mM) were analysed by CD spectroscopy to identify the secondary microstructure, respectively. Settings: scanning range of 180–260 nm, speed of 50 nm·min^−1^, bandwidth of 1 nm, and a response time of 0.5 s. Secondary structure proportions (α-helices, β-sheets, β-turns, and random coils) were analysed via CDNN v2.1 software (Applied Photophysics Ltd., Leatherhead, UK).

### 2.5. Proteinase Inhibition Assay

Proteolytic activity was assessed using β-casein (0.08 mM) as a substrate in a time-course assay (0–24 h). Enzymatic reactions were initiated by pre-incubating human MMP-1 with synthetic pHpG peptides (1:25 molar ratio) in a sodium phosphate buffer (pH 6.5) at 37 °C for 30 min. The recombinant human MMP-1 protein (rhMMP-1; ≥95% purity), obtained from MedChemExpress (Monmouth Junction, NJ, USA), was expressed in HEK293 cells as an unglycosylated form with a determined molecular weight of 49–61 kDa. The latent proenzyme requires activation via incubation with 1 mM aminophenylmercuric acetate at 37 °C for 2 h to generate the catalytically active form. Reactions were quenched with 5 mM EDTA at designated intervals, followed by electrophoretic separation on 12% SDS-PAGE and Coomassie staining. The negative control was β-casein in the absence of enzymatic catalysis, while water served as a system blank to monitor background interference.

### 2.6. Spectroscopic Characterisation

Ultraviolet–visible (UV–Vis) spectrum of pHpGs and zinc–pHpG complexes (0.1 mM) were recorded using a Shimadzu UV-2450 spectrophotometer (Shimadzu Corporation, Kyoto, Japan) in the wavelength range of 200–300 nm at room temperature.

Fourier-Transform infrared (FTIR) spectrum of pHpGs and zinc–pHpG complexes were acquired using a Spectra Nicolet iS50 FTIR spectroscopy (Thermo Fisher Scientific, Altrincham, UK). Lyophilised pHpGs were reconstituted to a 1 mM solution, incubated with zinc salts at 25 °C for 30 min to form a zinc–pHpG complex, and then subjected to ultrafiltration to remove unbound metal ions. Post-lyophilised peptides and complexes were determined by FTIR spectroscopy using KBr pellet methodology, and analysed in the range of 4000–400 cm^−1^ at room temperature.

### 2.7. Structure Refinement and Molecular Modelling

The three-dimensional visualisations of pHpG peptide’s structure were obtained using PEP-FOLD3 server (https://bioserv.rpbs.univ-paris-diderot.fr/services/PEP-FOLD3/). The structural template of MMP1 in the PDB format were obtained from the Protein Data Bank (https://www.rcsb.org/). Molecular docking between pHpGs and MMP1 was performed using the HDOCK server (http://hdock.phys.hust.edu.cn/, accessed on 11 March 2024) and visualised by PyMOL (2.5.4).

### 2.8. Scratch Wound Healing Assay

Human umbilical vein endothelial cells (HUVECs) were procured from the Shanghai Institute of Biochemistry and Cell Biology (Chinese Academy of Sciences, China). For the scratch wound healing assay, HUVECs were seeded in 24-well plates at a density of 1.2 × 10⁵ cells/well and cultured to form confluent monolayers. A linear wound was created by scraping with a sterile pipette tip. Serum-free Dulbecco’s modified Eagle medium containing 25 ng/mL of human vascular endothelial growth factor 165 (hVEGF165) was supplied, followed by the addition of either the pHpG-H7 peptide (3.125 μM, 6.25 μM, 12.5 μM, 25 μM) or water as a control. Cells were incubated for 12 h (37 °C, 5% CO_2_), and images were captured at 0 h (initial wound area) and 12 h. Wound healing was quantified using ImageJ 1.53t software, with a migration rate calculated as: Mobility (%) = (initial wound area–12 h residual area)/initial wound area × 100%.

### 2.9. Trans-Well Migration Assay

HUVECs (2 × 10^5^ cells) were seeded in the upper chamber of a modified Boyden chamber system (BD Sciences, Milpitas, CA, USA). The lower chamber contained medium with 10% FBS, hVEGF165 (25 ng/mL), pHpG-H7 peptide (3.125 μM, 6.25 μM, 12.5 μM, 25 μM), or water. After 24 h incubation (37 °C, 5% CO_2_), migrated cells on the lower membrane surface were fixed, stained with Crystal Violet, and imaged. Cell migration was quantified using ImageJ software.

### 2.10. Tube Formation Assay

Matrigel™ Basement Membrane Matrix (BD Bioscience, Franklin Lakes, NJ, USA) was added to a 96-well plate to a total volume of 60 μL per well and polymerised at 37 °C with 5% CO_2_ for 30 min to form a gel layer. HUVECs (2 × 10^4^ cells/well) in the medium with 1% FBS were seeded. Different concentrations of pHpG-H7 peptide (3.125 μM, 6.25 μM, 12.5 μM, 25 μM) were added to each well and incubated for 8 h (37 °C, 5% CO_2_). Results were detected by phase-contrast microscope (Olympus, Tokyo, Japan) and photographed. At least three representative fields from each well were selected to count the number of branching points.

### 2.11. Statistical Analysis

Data are presented as the mean ± SD of the results from at least three independent experiments. Comparisons were made using the Student’s *t*-test or one-way ANOVA according to the data type. Statistical analysis was performed using SPSS version 23.0 (IBM, Armonk, NY, USA). *p*-value < 0.05 was considered statistically significant.

## 3. Results

### 3.1. Two Novel pHpG Peptides Were Identified from A. squamigera Venom

Two novel pHpG peptides, pHpG-H5 (EDDH(5)GVG(10)) and pHpG-H7 (EDDH(7)GVG(10)), were identified from the lyophilised venom of *A. squamigera* (Figure 1A,B), which was encoded by two distinct cDNA clones. For comparative analysis, the previously characterised pHpG-H9 (EDDH(9)GVG(10)) was included as a reference for subsequent research to investigate the relationship between pHpG peptides [4].

Primary structures of three pHpG peptides—pHpG-H5, pHpG-H7, and pHpG-H9—served as templates for chemical synthesis (Table 1). The products were purified via reverse-phase HPLC and achieved a high degree of purity (>90%) as quantified by peak area percentage analysis (Figure 2). Molecular weights of the three synthetic peptides were determined via double-charged (z = 2) and triple-charged (z = 3) ions, as shown in Figure 3A–C, which matched the expected molecular weights based on their amino acid sequences (Table 1). Additional physicochemical properties, including theoretical isoelectric points (pI) and total average hydrophilicity, are summarised in Table 1.

### 3.2. pHpG Peptides Inhibit Metalloproteinase Activity via Zinc Ion Coordination

Building on previous findings that pHpG-H9 inhibits metalloproteinase-mediated casein hydrolysis [10], we investigated whether pHpG-H5 and pHpG-H7 exhibit similar inhibitory effects. Time-dependent β-casein degradation mediated by MMP-1 was analysed. In the absence of pHpG peptides, β-casein was progressively degraded and fully hydrolysed within 24 h (Figure 4A). In contrast, pre-incubation with pHpG-H5, pHpG-H7, or pHpG-H9 significantly inhibited MMP-1 activity in hydrolysing β-casein (Figure 4B–D). Notably, pHpG-H7 demonstrated the most pronounced inhibitory effect, suppressing MMP-1 activity as early as 0.5 h.

Given the critical importance of zinc ions in the catalytic function of metalloproteinases, we investigated the metal-chelating properties of pHpG peptides. All three peptides demonstrated the strong affinity of zinc ions, with optimal binding occurring at a molar ratio of 1:100 (Figure 5A–C). Significant shifts in the zinc binding were observed by UV–visible spectroscopy: from 214 nm to 216 nm for pHpG-H5, from 212 nm to 215 nm for pHpG-H7, and from 210 nm to 218 nm for pHpG-H9 (Figure 5D–F).

Conformational changes were monitored by FTIR spectroscopy to characterise the functional groups involved in zinc chelation. All pHpG peptides exhibited typical protein infrared spectrum, with amide I bands (C=O stretching vibrations) in the range of 1600–1700 cm^−1^ and amide II bands (N–H bending and C–N stretching vibrations) in the range of 1500–1600 cm^−1^. Peaks within the amide I region suggest the presence of α-helices and unordered coils [11]. Upon zinc binding, red-shifts (shifts to lower wavenumbers) were detected in both the amide I and amide II regions (Figure 5G–I), indicative of conformational changes in the zinc–pHpG complexes.

### 3.3. Understanding Interaction Mechanism of pHpGs by Structural Analysis and Molecular Modelling

The structural characterisation of pHpG peptides was employed by circular dichroism spectroscopy. The CD spectrum of all three pHpGs exhibit a strong negative band in the range of 190–200 nm and a positive band above 210 nm (Figure 6), indicating the presence of predominantly coils or disordered regions, as previously reported [12]. Further analysis reveals the following secondary structure distributions: 12.1% α-helix, 40.6% β-strand, 21.6% β-turn, and 29.2% random coil for pHpG-H5; 6.2% α-helix, 13.6% β-strand, 29.4% β-turn, and 46.4% random coil for pHpG-H7; 6.3% α-helix, 11.7% β-strand, 30.7% β-turn, and 47.5% random coil for pHpG-H9 (Table 2). These findings are supported by the conformational landscape conducted using the PEP-FOLD3 software. The structure prediction profiles for all three pHpG peptides (Figure 7A–C) display predominantly coil-dominated secondary structures, which are favourable for zinc binding and facilitate chelation between the polypeptides and zinc ions [13].

To further investigate the interaction mechanism between pHpG peptides and MMP-1, a 3D model of MMP-1 was constructed and subjected to molecular docking studies. The potential interaction sites of pHpG peptides with MMP-1 are shown in Figure 8A–C, overlapping with the Zn^2+^-binding fragment. The amino acids involved in coordination between pHpG-H5, pHpG-H7, pHpG-H9, and MMP-1 were detailed in Figure 8D, as shown in Figure 8E and Figure 8F, respectively, suggesting that the histidine residues in pHpG peptides disrupt zinc-binding within the central catalytic domain of MMP-1.

### 3.4. Pharmacological Implications of pHpG-H7: Anti-Cell Migration and Anti-Angiogenesis

Given the pivotal role of MMPs in pathophysiological processes such as angiogenesis and wound healing, pHpG-H7—which demonstrated the most significant inhibitory effect on MMP-1—was selected for subsequent investigation. The in vitro wound healing and Trans-well cell migration assay of HUVECs were employed. As illustrated in Figure 9A, the wound closure capability of HUVECs was stimulated by hVEGF165, but significantly impaired by the treatment with pHpG-H7 across all groups (*p* < 0.01 for 3.125 and 6.25 μM, *p* < 0.0001 for 12.5 and 25 μM). Consistent results have been illustrated in the cell migration assay (Figure 9B), where the HUVECs migration was inhibited by pHpG-H7 in a dose-dependent manner (*p* < 0.001 for 3.125 and 6.25 μM, *p* < 0.0001 for 12.5 and 25 μM). Moreover, pHpG-H7 demonstrate potential anti-angiogenic activity in vitro. As depicted in Figure 10A,B, compared to the control group, hVEGF165 stimulation significantly increased the number of blood vessels; however, this effect is attenuated by pHpG-H7 treatment. Statistically significant differences are observed among the groups treated with pHpG-H7 at increasing concentrations (*p* < 0.01 for 3.125 μM, *p* < 0.0001 for 6.25, 12.5, and 25 μM), indicating that pHpG-H7 exerts anti-angiogenesis effects in a dose-dependent manner.

## 4. Discussion

In our study, the precursors of pHpG-H5 and pHpG-H7 peptides were cloned from a cDNA library of *A. squamigera*, exhibiting differences in histidine and glycine residue composition and alignment. Combined with previous research [4], these results indicate the existence of multiple isoforms of pHpG peptides in *A. squamigera* (Table 3). The identification of peptide isoforms within a single species suggests a high degree of sequence diversity, potentially reflecting the functional specialisation or evolutionary adaptation of the vipers [14,15,16].

In the preceding investigation, we reported the presence of snake venom metalloproteinase in *A. squamigera*’s venom [3], which belongs to the Pep_M12B_propeptide family. Based on the findings of Wagstaff et al., Atheris pHpG-1 (equivalent to pHpG-H9 in this work) revealed the alleviation of venom-induced haemorrhage and the inhibition of SVMP activity [10]. Interestingly, considering the co-existence of SVMPs and pHpG peptides in the same venom, pHpG peptides may act as the endogenous inhibitors of metalloproteinases within *Atheris* species. This hypothesis is strongly supported by findings in numerous viper species regarding the presence of endogenous inhibitors of snake venom enzymes [17,18]. The post-envenoming administration of metalloproteinase inhibitors could effectively reduce local haemorrhage and systemic lethality, demonstrating the therapeutic potential of targeted protease inhibition as an adjunct strategy for viper envenomation [19,20].

Proteinase inhibition assay of MMP-1 revealed potent catalytic suppression by all pHpG peptides from *A. squamigera* (pHpG-H5, pHpG-H7, and pHpG-H9). The binding of the pHpG peptides to Zn^2+^ was determined. UV–vis and infrared spectroscopy analyses indicate conformational alterations in the zinc–pHpG complexes across all three pHpG peptides, in which the intrinsically disordered regions enable dynamic metal ion interactions via conformational adaptability [21]. In polypeptides, histidine residues are likely to play a role in chelating metal ions and participate in substrate binding, while glycine-rich motifs reduce steric constraints, enhancing metal-binding accessibility [6,22]. Molecular modelling in our study supports this mechanism: the histidine residues of pHpG peptides selectively bind to the Zn^2+^ cofactor of MMP-1 by the non-covalent bond (Figure 9; His-6 for pHpG-H5, His-7 for pHpG-H7 and His-7 for pHpG-H9), and glycine does not directly engage in the coordination process. This highlights the potential mechanism of pHpG peptides as zinc-chelating inhibitors capable of impairing metalloprotease activity.

As fundamental regulators of tissue remodelling and wound healing, matrix metalloproteinases play a crucial role in maintaining physiological balance; their dysregulation is closely associated with numerous pathological conditions, including cardiovascular diseases, cancer, and chronic inflammation [23,24]. Thus, MMP inhibitors (MMPIs) hold significant therapeutic potential for modulating these processes [25,26]. In our study, pHpG-H7 displayed anti-cell migration and anti-angiogenic activities in vitro, suggesting its potential to influence processes such as wound healing and tumour progression. These findings underscore the broader implications of pHpG peptides as multi-functional inhibitors with diverse biological activities.

In this work, pHpG peptide exhibited bioactivity at micromolar-range concentrations (3.2–25 µM) in vitro, aligning with previous studies in which analogous phenomena were empirically validated and systematically elucidated [27]. The administration of candidates with micromolar-range concentration carries risks of cytotoxicity and dose-limiting adverse effects. In contrast, snake venom components potentiate bioactivity via stereoselective molecular targeting and envenomation-driven spatial enrichment in biological niches. This reveals biological complexity and adaptability, highlighting nature-derived insights for peptide-based drug discovery.

The identification of pHpG peptides as potent MMPIs opens new avenues for their application in drug discovery, offering promising opportunities for the development of novel therapeutics. Although there are challenges in peptide drug development and the formulation of drug delivery systems, peptide-based therapeutics remain compelling candidates due to their unique pharmacological advantages. Unlike traditional small molecule inhibitors, which often suffer from poor selectivity and systemic toxicity, pHpG peptides offer a natural alternative with inherent specificity and biocompatibility [28,29]. Their ability to chelate the central Zn^2+^ ion in the active site of MMPs provides a mechanistic basis for their inhibitory activity, and their structural diversity suggests the possibility of targeting multiple MMP isoforms or other metalloenzymes [30,31]. Additionally, the modular layout of pHpG peptides, characterised by alternating histidine and glycine residues, enables rational design strategies to optimise potency, stability, and pharmacokinetic properties [32]. The further elucidation of the pharmacological activities, cytotoxic profiles, and molecular mechanisms could facilitate the design of MMPIs targeting zinc-dependent processes, thereby improving therapeutic strategies for related diseases.

## 5. Conclusions

The pHpG peptides isolated from snake venom of *Atheris squamigera* represent a promising class of natural inhibitors with potential applications in MMP-related therapies. The exceptional structural diversity of pHpG peptides, combined with their zinc-dependent mechanism of action, positions them as promising candidates for the development of novel MMPIs.

## Figures and Tables

**Figure 1 biomolecules-15-00706-f001:**
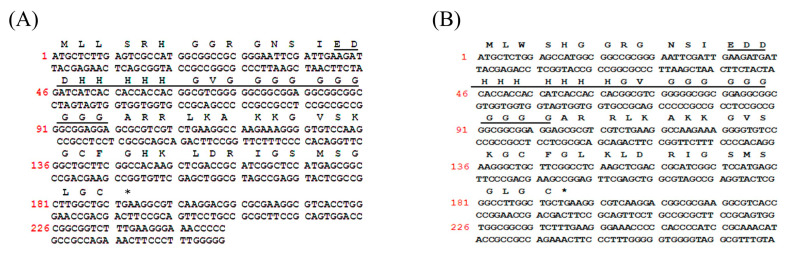
Nucleotide and deduced amino acid sequences of open reading frame encoding precursors: (**A**) pHpG-H5 precursor and (**B**) pHpG-H7 precursor. Sequences of the mature peptide were underlined; stop codons were denoted by asterisks.

**Figure 2 biomolecules-15-00706-f002:**
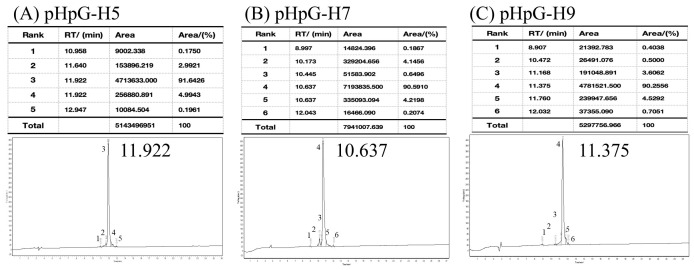
Reverse phase-HPLC of synthetic peptides demonstrate high purity: (**A**) pHpG-H5, (**B**) pHpG-H7, (**C**) pHpG-H9. RT: Retention time; Area: Peak area; Area/(%): Peak area percentage.

**Figure 3 biomolecules-15-00706-f003:**
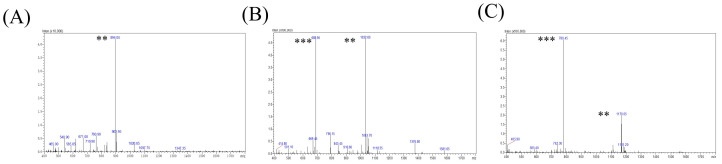
Mass spectrometry characterisation of synthetic pHpG peptides with doubly charged and triply charged ions: (**A**) pHpG-H5; (**B**) pHpG-H7; and (**C**) pHpG-H9. The asterisk-based notation serves to indicate the charge states of molecular ions ** doubly-charged (*z* = 2), *** triply-charged (*z* = 3).

**Figure 4 biomolecules-15-00706-f004:**
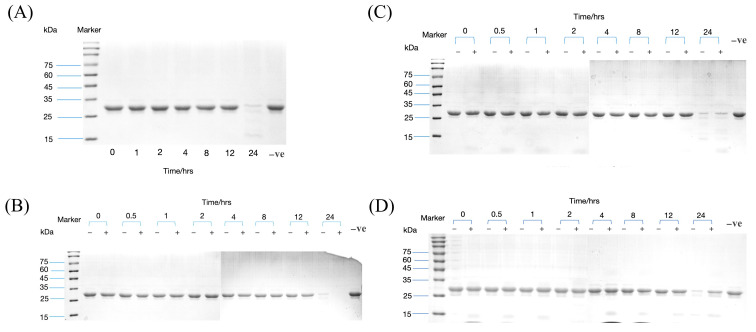
MMP-1 inhibition assay of synthetic pHpG peptides. (**A**) Time-dependent β-casein degradation mediated by MMP-1. Time-dependent β-casein degradation mediated by MMP-1 pre-incubated with pHpG peptides: (**B**) pHpG-H5, (**C**) pHpG-H7, (**D**) pHpG-H9. Molecular mass markers (kDa) in lane M and negative control (vehicle) in the lane-ve. Original SDS-PAGE images can be found in Appendix A.

**Figure 5 biomolecules-15-00706-f005:**
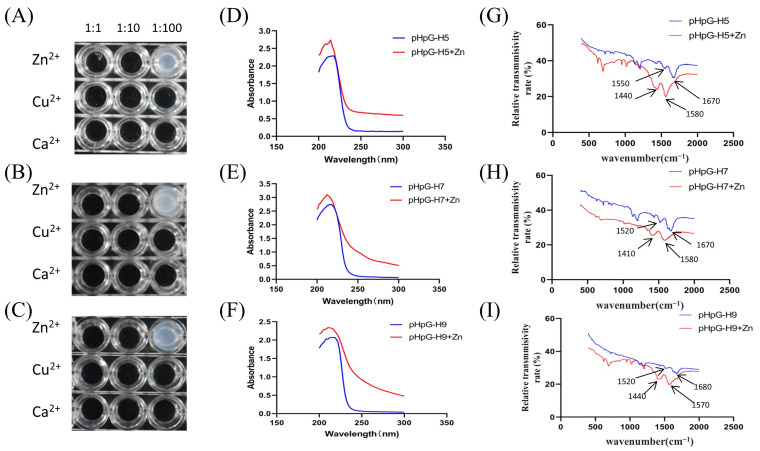
Zinc–ion coordination properties of the pHpG peptides. Metal-cheating abilities (Zn^2+^, Cu^2+^, Ca^2+^) of pHpG peptides were detected at molar ratios of 1:1, 1:10, and 1:100, (**A**) pHpG-H5, (**B**) pHpG-H7, (**C**) pHpG-H9. UV–Vis spectroscopic determination, (**D**) pHpG-H5 peptide and Zn^2+^-pHpG-H5 complex, (**E**) pHpG-H7 and Zn^2+^-pHpG-H7 complex, (**F**) pHpG-H9 and Zn^2+^-pHpG-H9 complex. FTIR spectroscopic determination: (**G**) pHpG-H5 peptide and Zn^2+^-pHpG-H5 complex, (H) pHpG-H7 and Zn^2+^-pHpG-H7 complex, (**I**) pHpG-H9 and Zn^2+^-pHpG-H9 complex.

**Figure 6 biomolecules-15-00706-f006:**
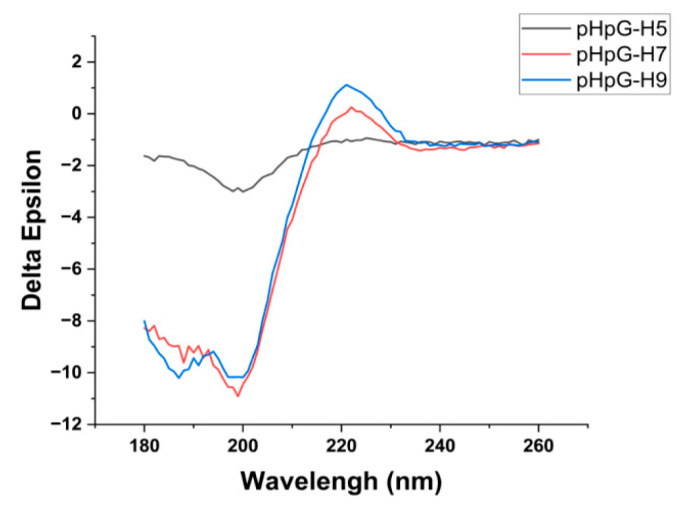
Circular dichroism spectroscopy analyses of pHpG peptides. CD spectrum (190–260 nm) of pHpG-H5, pHpG-H7, and pHpG-H9 were annotated separately.

**Figure 7 biomolecules-15-00706-f007:**
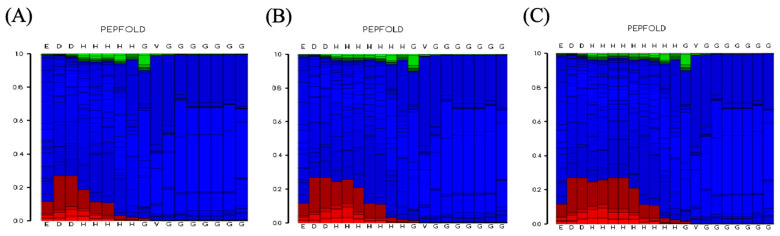
Conformational landscape of pHpG peptides. Rosetta-based structure prediction profiles of pHpGs: (**A**) pHpG-H5, (**B**) pHpG-H7, (**C**) pHpG-H9. Vertical axis: energy score (0–1.0); horizontal axis: Position of amino acid residues. Colour-coded probabilities: red (α-helices), blue (random coils), green (β-strands).

**Figure 8 biomolecules-15-00706-f008:**
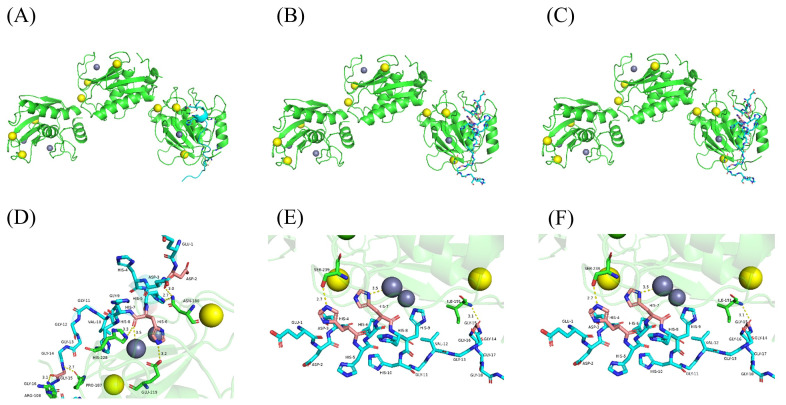
Schematic for the docked conformation of MMP-1 with pHpG peptides: (**A**) pHpG-H5; (**B**) pHpG-H7; and (**C**) pHpG-H9. The HDOCK-generated structural model of MMP-1 and pHpG peptides was visualised in detail, highlighting the binding site and docking interactions between pHpGs and MMP-1, respectively: (**D**) pHpG-H5, (**E**) pHpG-H7, and (**F**) pHpG-H9.

**Figure 9 biomolecules-15-00706-f009:**
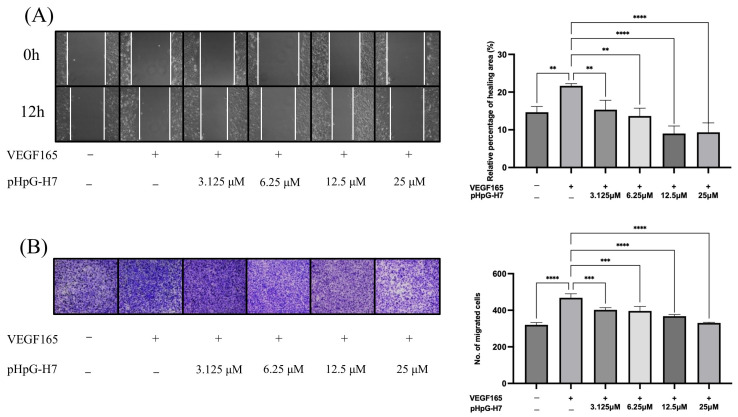
pHpG-H7 inhibits the cell migration of HUVECs in vitro. (**A**) wound healing assay of hVEGF165-stimulated HUVECs treated with pHpG-H7 (3.125 μM, 6.25 μM, 12.5 μM, 25 μM) for 12 h; ** *p* < 0.01, **** *p* < 0.0001. (**B**) Trans-well assay was used to explore cell migration ability of hVEGF165-stimulated HUVECs treated with pHpG-H7 (3.125 μM, 6.25 μM, 2.5 μM, 25 μM) for 24 h; *** *p* < 0.001 **** *p* < 0.0001.

**Figure 10 biomolecules-15-00706-f010:**
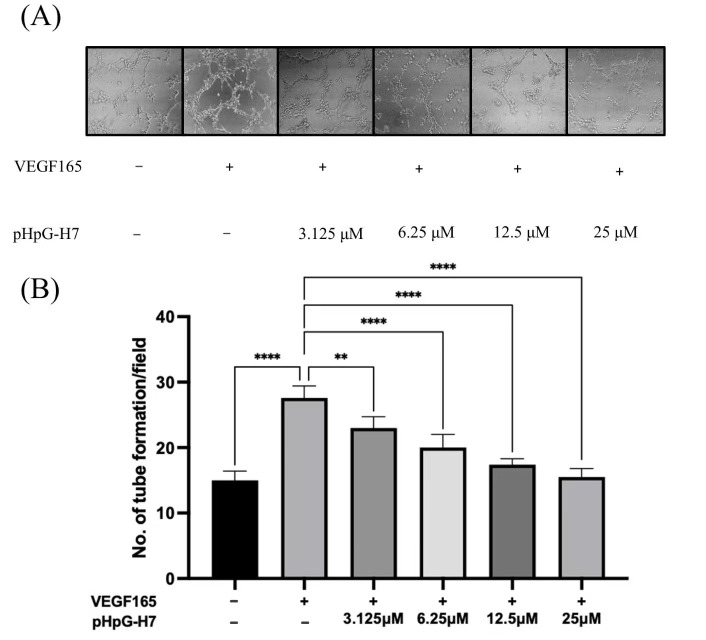
hVEGF165-induced endothelial angiogenesis in Matrigel. (**A**) Phase-contrast imaging of tubular network formation on pHpG-H7-treated Matrigel (3.125 μM, 6.25 μM, 12.5 μM, 25 μM). (**B**) Quantification of branching points (≥50 μm length) per field (8 h incubation); ** *p* < 0.01, **** *p* < 0.0001.

**Table 1 biomolecules-15-00706-t001:** Comparison of three pHpG peptides identified from *A. squamigera*’s venom.

Name	Full Sequence	Abb Seq	Ave Hyd	Calculated MW/(Da)	Observed MW/(Da)	pI
pHpG-H5	EDDHHHHHGVGGGGGGGGGG	EDDH_(5)_GVG_(10)_	0.3	1789.72	1789.73 ^a^	5.8
					894.05 **	
pHpG-H7	EDDHHHHHHHGVGGGGGGGGGG	EDDH_(7)_GVG_(10)_	0.2	2064.00	2064.01 ^a^	6.1
					1032.60 **	
					688.95 ***	
pHpG-H9	EDDHHHHHHHHHGVGGGGGGGGGG	EDDH_(9)_GVG_(10)_	0.1	2338.28	2338.30 ^a^	6.3
					1170.05 **	
					780.45 ***	

(Abb seq: abbreviation sequence; Ave Hyd: average hydrophilicity; MW: molecular weight; pI: isoelectric point; ^a^ deduced from double- and triple-charged ions; ** *z* = 2; *** *z* = 3).

**Table 2 biomolecules-15-00706-t002:** Secondary structure composition of pHpG peptides analysed by CDNN software.

Temperature (°C)	pHpGs	α-Helix (%)	β-Strand (%)	β-Turn (%)	Unorderd (%)	Total (%)
20	pHpG-H5	12.1	40.6	21.6	29.2	103.5
	pHpG-H7	6.2	13.6	29.4	46.4	95.5
	pHpG-H9	6.3	11.7	30.7	47.5	96.2

**Table 3 biomolecules-15-00706-t003:** Structural and post-translational variants of pHpG peptides in *Atheris squamigera*.

Sample Source	Deduced Peptide Sequence	Significance	Attribution Method
*Atheris squamigera* venom	EDDH(7)GVG(10)…	***	Molecular cloning ^a^
	EDDH(9)GVG(10)…	**	Molecular cloning ^a^;
		***	MS/MS [4]
	EDDH(5)GVG(10)…	**	Molecular cloning ^a^
	EDDH(10)GVG(10)…	*	MS/MS [4]
	EDDH(11)GVG(10)…	******	MS/MS [4]

^a^ Note: Molecular cloning of *Atheris squamigera* venom gland cDNA library in this study. The asterisk notation (*, **, ***) denotes significance, serving as an indicator of the reliability and reproducibility of experimental findings.

## Data Availability

The original contributions presented in this study are included in the article/Appendix A. Further inquiries can be directed to the corresponding author.

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
