# Peer review of "Discovery and Characterisation of Novel Poly-Histidine-Poly-Glycine Peptides as Matrix Metalloproteinase Inhibitors"

_biomolecules, 2025, doi:10.3390/biom15050706_

Round 1
Reviewer 1 Report
Comments and Suggestions for Authors
The article titled "Discovery and Characterization of Novel Poly-Histidine-Poly-Glycine Peptides as Matrix Metalloproteinase Inhibitors," submitted to the journal "Biomolecules," presents novel information on pHpG peptides identified in the venom of the snake Atheris squamigera. The Authors identified for the first time two poly-histidine-poly-glycine peptides (pHpG-H5 and pHpG-H7) as promising candidates for matrix metalloproteinase inhibitors. That is why I consider the topic as original and relevant to the field as it adds new information on understanding the biological mechanisms and potential therapeutic applications of the described inhibitors.
The studies were conducted using standard cell biology methods and chemical analyses. The creation of a gel matrix and the incubation of HUVECs with varying concentrations of pHpG-H5 and pHpG-H7 peptides yielded interesting results. The use of phase contrast microscopy to observe branching points, along with statistical analysis, demonstrates that the authors placed great emphasis on the reliability of their findings. I consider the methods used to be appropriate and sufficient to conduct the planned research at a high level.
The identified peptides, pHpG-H5 and pHpG-H7, exhibited a strong inhibitory effect on MMP-1 activity, as confirmed by the results of the β-casein degradation analysis. Surprisingly, pHpG-H7 emerged as the most effective inhibitor, potentially opening new avenues for research into therapies for diseases associated with MMP dysregulation.
The authors employed UV-Vis spectroscopy and FTIR to investigate the zinc ion chelating ability of the peptides. The results suggest that the pHpG peptides can interact with metals, which is crucial for their function as MMP inhibitors. Additionally, molecular modeling highlights potential interaction sites of the peptides with MMP-1, marking an important step toward elucidating their mechanism of action.
The finding that pHpG-H7 exhibits antiangiogenic and cell migration-inhibiting properties indicates that these peptides may have applications in cancer therapy and wound healing. While further studies are necessary to assess their efficacy and safety in clinical settings, the results presented in the article are promising.
Overall, the conclusions are consistent with the evidence and arguments presented and they address the main question. The article provides compelling evidence that pHpG peptides derived from Atheris squamigera venom represent a promising class of natural MMP inhibitors. Their unique structure and mechanism of action, based on zinc chelation, may contribute to the development of new therapies for diseases in which MMPs play a crucial role. Continued research in this area is warranted to fully explore the therapeutic potential of these peptides in medicine.
Cited references are appropriate, excessive self-citation is not observed.
Minor comments: Please add the missing units in figures and tables, such as for concentration and molecular weight.
Author Response
Comments 1:
Please add the missing units in figures and tables, such as for concentration and molecular weight.
Response 1:
We sincerely appreciate your valuable suggestions. The missing units in the figures and tables, including concentration and molecular weight, have been fully modified and supplemented in the manuscript as recommended.
Reviewer 2 Report
Comments and Suggestions for Authors
The study by Wang and Colleagues contains interesting work about the presence of MMP-inhibitory peptides with oligo-histidine sequences in specific snake venom proteomes. The authors used a variety of methods to provide experimental evidences for their statements. Starting form cDNA-derived sequences, they produced and critically analysed synthetic peptides, and therefafter used these in enzyme inhibition and in biological activity tests. A number of elements may contribute to enhance this study to a higher scientific level and to broaden the readership for this work.
Major Comments
- My judging of the quality of research is not only based on the experimental data, but also on the quality of the sections Materials & Methods, Discussion and knowledge of literature. Whereas, the writing is well done, a number of flaws need attention. In the M&M section, we read: " molecular cloning was from 5 mg venom. This is, of course, nonsense, because snake venoms are proteinaceous secretomes from glands and hardly contain cells, from which mRNAs may be isolated for cDNA cloning. Happily and to some rescue of this manuscript, one reads in the Results section that the cDNAs were "from a venom-derived library". Even this statement is odd, because it should be "from a venom-secreting gland- or cell-derived library". When reading such sentences, one wonders whether the experiment has been really done. Therefore, these oddities need clear reformulations. The authors need to clarify how the cDNA library was made (which tissue?, which cells?, self-made? or commercially available cDNA libray?) and how the cDNAs encoding these peptides were identified (selection procedure or just by blunt sequencing, sources of information, etc.). At present, not any explanation is provided how the discovery of these poly-histidine, poly-glycine peptides was made. Was it just by analogy with the previously mentioned study or in a different way?
- In the M&M section a proteinase inhibition assay is explained with beta-casein as substrate of MMP-1. Which MMP-1 was used? Was it human, mouse or other MMP-1? Was it recombinant and in which expression system, was it glycosylated? Was it expressed in its pro-form and if yes, how was it activated? Was it made in-house or commercially procured? You may wonder why all these questions are raised. These are brought up because researchers within the fields of MMPs care about these things, in particular if a study (including the present one) has the potential to raise their attention.
- In the Results section about the (human?) MMP-1 inhibition assay a gel-based system was used to indicate proteolysis of beta-casein by casein staining analysis. In relation to the follow-up biological data (e.g. HUVEC scratch assay with the use of VEGF, matrigel/VEGF test), no dose-response quantitative data are provided about the MMP-1 inhibition. The authors need to provide these enzyme inhibition data in a proper way, because this test is the only DIRECT test of enzyme inhibition. In other words, the VEGF-induced tests with endothelial cells are surrogate tests of biological activity and do not prove that this is by proteinase inhibition. Alternatively and as suggested here, the authors could use a commercially available quenched fluorigenic substrate of MMP-1 and test their synthetic peptides for inhibitory activity with clear outcomes (Michaelis Menten kinetic data, type of inhibition such as competitive versus non-competitive inhibition). Such data would also be relevant in relation to the Discussion about protential applications of such peptides.
- The authors document metal chelating activities of the synthetic peptides and inhibition of VEGF-induced endothelial cell migration in various assays. In these assays the inhibitory activity is in the µM range (3.2 to 25 µM), which is rather high and would need doses that are extremely high to abtain in vivo activity. Furthermore, such peptides wouldhave to be given parenterally (are broken down after peroral administration). On the other hand, these elements are completely different within the snake biology. Please try to explain this point in a more and extended way,
- The fact that oligohistidine inhibits MMP activity is not new. More than 20 years ago, we studied an oligo-histidine-tagged antibody derivative as inhibitor of MMP-9 and used as a control a polyhistidine. We discovered that our control preparation polyhistidine inhibits MMPs by itself. Please read the following manuscript carefully [Biochem Pharmacol 2004 Mar 1;67(5):1001-9. doi: 10.1016/j.bcp.2003.10.030. Inhibitors of gelatinase B/matrix metalloproteinase-9 activity comparison of a peptidomimetic and polyhistidine with single-chain derivatives of a neutralizing monoclonal antibody. Jialiang Hu et al.]. The combination of your data with the above-mentioned prior data indicates and inforces the notion that it is poly-histidine (and not the combination of poly-histidine with poly-glycine) which is inhibitory for MMP-1 (your study) and MMP-9 and MMP-2 (our study). In addition, the inhibition of MMP-9 and MMP-2 was also in the micromolar range and is rather low in comparison commercially available pharmaceuticals (Nature Reviews Drug Discovery 2007 DOI 10.038/nrd2308).
Author Response
Comments 1: [My judging of the quality of research is not only based on the experimental data, but also on the quality of the sections Materials & Methods, Discussion and knowledge of literature. Whereas, the writing is well done, a number of flaws need attention. In the M&M section, we read: " molecular cloning was from 5 mg venom. This is, of course, nonsense, because snake venoms are proteinaceous secretomes from glands and hardly contain cells, from which mRNAs may be isolated for cDNA cloning. Happily and to some rescue of this manuscript, one reads in the Results section that the cDNAs were "from a venom-derived library". Even this statement is odd, because it should be "from a venom-secreting gland- or cell-derived library". When reading such sentences, one wonders whether the experiment has been really done. Therefore, these oddities need clear reformulations. The authors need to clarify how the cDNA library was made (which tissue?, which cells?, self-made? or commercially available cDNA libray?) and how the cDNAs encoding these peptides were identified (selection procedure or just by blunt sequencing, sources of information, etc.). At present, not any explanation is provided how the discovery of these poly-histidine, poly-glycine peptides was made. Was it just by analogy with the previously mentioned study or in a different way?]
Response 1: [We sincerely appreciate your valuable comments. As you mentioned, snake venom consists mainly of proteins and polypeptides, but may contain trace amounts of cellular debris, including shed venom gland epithelial cells. On the one hand, we used commercially obtained venom collected by manual milking, a method that increases the cellular contamination. On the other hand, we chose a highly sensitive RNA extraction method via oligo(dT) magnetic beads to confirm mRNA enrichment. In our study, the 3'-RACE procedure was employed to obtain full-length nucleic acid sequence data using a SMART-RACE kit. The specific sense primer was designed to target a highly conserved domain of the 5' untranslated region of previously characterised pHpG peptide from related snake species. NCBI BLAST searches were employed to identify the target peptides based on homology alignment. This approach is supported by a series of publications on molecular cloning of target cDNAs from snake venom and frog skin secretions.(DOI: 10.1016/j.peptides.2010.09.023; DOI: 10.1016/j.toxicon.2013.05.012; DOI: 10.3390/toxins8060168,Molecular Characterization of Three Novel Phospholipase Aâ‚‚ Proteins from the Venom of Atheris chlorechis, Atheris nitschei and Atheris squamigera)]
Comments 2: [In the M&M section a proteinase inhibition assay is explained with beta-casein as substrate of MMP-1. Which MMP-1 was used? Was it human, mouse or other MMP-1? Was it recombinant and in which expression system, was it glycosylated? Was it expressed in its pro-form and if yes, how was it activated? Was it made in-house or commercially procured? You may wonder why all these questions are raised. These are brought up because researchers within the fields of MMPs care about these things, in particular if a study (including the present one) has the potential to raise their attention. ]
Response 2 : [Thank you very much for your valuable recommendations. In protease inhibition assay, a recombination human matrix metalloproteinase-1 (rhMMP-1, EC 3.4.24.7) was employed as the enzymatic catalyst, with type I collagen serving as the physiological substrate in accordance with established method. The experimental details are summarised as follows: a recombinant human MMP-1 protein (≥95% purity) was obtained from MedChemExpress (Monmouth Junction, NJ, USA). This unglycosylated protein was heterologously expressed in HEK293 cells, and its molecular weight was determined to be 49-61 kDa by SDS-PAGE analysis under reducing conditions. The latent proenzyme requires activation via incubation with 1 mM aminophenylmercuric acetate (APMA) at 37°C for 2 hours to generate the catalytically active form. As recommended, we have rigorously revised the manuscript.]
Comments 3: [In the Results section about the (human?) MMP-1 inhibition assay a gel-based system was used to indicate proteolysis of beta-casein by casein staining analysis. In relation to the follow-up biological data (e.g. HUVEC scratch assay with the use of VEGF, matrigel/VEGF test), no dose-response quantitative data are provided about the MMP-1 inhibition. The authors need to provide these enzyme inhibition data in a proper way, because this test is the only DIRECT test of enzyme inhibition. In other words, the VEGF-induced tests with endothelial cells are surrogate tests of biological activity and do not prove that this is by proteinase inhibition. Alternatively and as suggested here, the authors could use a commercially available quenched fluorigenic substrate of MMP-1 and test their synthetic peptides for inhibitory activity with clear outcomes (Michaelis Menten kinetic data, type of inhibition such as competitive versus non-competitive inhibition). Such data would also be relevant in relation to the Discussion about protential applications of such peptides.]
Response 3: [We sincerely appreciate your constructive feedback. In this work, we aim to present the identification of novel pHpG peptides through molecular cloning, proteinase inhibition assay, and investigations into other pharmacological effects. In accordance with your recommendation, we will utilise a fluorogenic substrate specific to human MMP-1 in subsequent investigations to quantitatively assess the inhibitory activity of synthesised peptides through systematic evaluation of enzyme activity modulation. This experimental approach will enable quantitative determination of Michaelis-Menten kinetic parameters through fluorimetric assays, followed by rigorous analysis to characterise the inhibition modality. Your expertise has been instrumental in elevating the technical validity and scientific clarity of our findings.]
Comments 4: [The authors document metal chelating activities of the synthetic peptides and inhibition of VEGF-induced endothelial cell migration in various assays. In these assays the inhibitory activity is in the µM range (3.2 to 25 µM), which is rather high and would need doses that are extremely high to abtain in vivo activity. Furthermore, such peptides wouldhave to be given parenterally (are broken down after peroral administration). On the other hand, these elements are completely different within the snake biology. Please try to explain this point in a more and extended way.]
Response 4: [We sincerely appreciate your insightful comments. As you mentioned, the inhibitory effect on endothelial cell migration was observed at micromolar concentrations in our study. The administration of relatively high doses may lead to potential toxicity and side-effects, and pose challenges in formulating suitable drug delivery systems. Conversely, snakes have evolved specialized physiological adaptations, whereby venom components (e.g., neurotoxins, cytotoxins) exhibit biological efficacy through target-specific molecular interactions. Furthermore, venom peptides exert potent pharmacological effects through localized high concentrations by envenomation. This highlights the complexity and adaptability of biological systems, suggesting that there is substantial knowledge to be gleaned from nature when developing novel peptide-based drugs. Thanks again for your expert recommendations and we have rigorously addressed revision in the manuscript.]
Comments 5: [The fact that oligohistidine inhibits MMP activity is not new. More than 20 years ago, we studied an oligo-histidine-tagged antibody derivative as inhibitor of MMP-9 and used as a control a polyhistidine. We discovered that our control preparation polyhistidine inhibits MMPs by itself. Please read the following manuscript carefully [Biochem Pharmacol 2004 Mar 1;67(5):1001-9. doi: 10.1016/j.bcp.2003.10.030. Inhibitors of gelatinase B/matrix metalloproteinase-9 activity comparison of a peptidomimetic and polyhistidine with single-chain derivatives of a neutralizing monoclonal antibody. Jialiang Hu et al.]. The combination of your data with the above-mentioned prior data indicates and inforces the notion that it is poly-histidine (and not the combination of poly-histidine with poly-glycine) which is inhibitory for MMP-1 (your study) and MMP-9 and MMP-2 (our study). In addition, the inhibition of MMP-9 and MMP-2 was also in the micromolar range and is rather low in comparison commercially available pharmaceuticals (Nature Reviews Drug Discovery 2007 DOI 10.038/nrd2308). ]
Response 5 : [Sincerely thanks for your insightful suggestions. The discussion section of this manuscript has been substantively expanded by incorporating the seminal literature you recommended, which significantly enhances the academic rigor of our arguments.]
Round 2
Reviewer 2 Report
Comments and Suggestions for Authors
The authors did a number of efforts to improve the quality of their research. Although they did not perform some suggested additional experiments, they clain to do that in forthcoming research. For the time being this is acceptable.
Still, a two clarifications/changes could improve and strengthen their work, as it is in the details that perfection resides.
1. It is indeed explained that the cDNA library is made with commercial reagents, but for any reader with molecular biology training the phrasing in the manuscript remains odd. As the authors state, the glandular cell contamination in snake venom (which is mainly a proteinaceous secretion product) is used to isolate mRNA and to reverse trascribe that mRNA into cDNA reverse transcription. For that reason, it is preferable that the authors would write.
"A commercial cDNA library was generated by reverse transcription of mRNA from glandular cells present in snake venom......."
2. The authors have indeed now added the earlier finding that polyhistidine inhibits MMPs. However, they just copied the (INCORRECT!) Pubmed citation and did not do the effort to look at and cite the original paper correctly. Indeed, the punctuation of the title and subtitle of the Hu et al. paper is as follows and thus needs correction into two sentences with punctuation.
"Inhibitors of gelatinase B/matrix metalloproteinase-9 activity: Comparison of a peptidomimetic and polyhistidine with single-chain derivatives of a neutralizing monoclonal antibodies"
With kind regards and congratulations with this interesting research
Ghislain Opdenakker
Author Response
Comment 1: [It is indeed explained that the cDNA library is made with commercial reagents, but for any reader with molecular biology training the phrasing in the manuscript remains odd. As the authors state, the glandular cell contamination in snake venom (which is mainly a proteinaceous secretion product) is used to isolate mRNA and to reverse trascribe that mRNA into cDNA reverse transcription. For that reason, it is preferable that the authors would write.
"A commercial cDNA library was generated by reverse transcription of mRNA from glandular cells present in snake venom......."]
Response 1: [Thanks again for your expert recommendations. We have rigorously addressed revision in the manuscript. Your suggestions provided critical insights that elucidate the methodology employed in this study, thereby preventing potential confusion among readers. ]”
Comments 2: [The authors have indeed now added the earlier finding that polyhistidine inhibits MMPs. However, they just copied the (INCORRECT!) Pubmed citation and did not do the effort to look at and cite the original paper correctly. Indeed, the punctuation of the title and subtitle of the Hu et al. paper is as follows and thus needs correction into two sentences with punctuation.
"Inhibitors of gelatinase B/matrix metalloproteinase-9 activity: Comparison of a peptidomimetic and polyhistidine with single-chain derivatives of a neutralizing monoclonal antibodies" ]
Response 2: [We sincerely apologize for the oversight in citing the paper of Hu et al.. Your guidance on the significance of accurate referencing of the original source is greatly appreciated. We have meticulously revisited the original publication. The correct title punctuation has been verified, and we've revised the citation in the reference list to rectify this error. Moreover, all other citations in the manuscript have been checked again. We truly value the thorough review you've conducted and deeply regret any inconvenience this error may have caused. ]”